# How Do Cartilage Lubrication Mechanisms Fail in Osteoarthritis? A Comprehensive Review

**DOI:** 10.3390/bioengineering11060541

**Published:** 2024-05-24

**Authors:** Manoj Rajankunte Mahadeshwara, Maisoon Al-Jawad, Richard M. Hall, Hemant Pandit, Reem El-Gendy, Michael Bryant

**Affiliations:** 1Institute of Functional Surfaces, Mechanical Engineering, University of Leeds, Leeds LS2 9JT, UK; 2Department of Oral Biology, Faculty of Dentistry, University of Leeds, Leeds LS2 9JT, UK; m.al-jawad@leeds.ac.uk (M.A.-J.); r.el-gendy@leeds.ac.uk (R.E.-G.); 3School of Engineering, College of Engineering and Physical Sciences, University of Birmingham, Birmingham B15 2TT, UK; r.m.hall@bham.ac.uk; 4Leeds Institute of Rheumatic and Musculoskeletal Medicine (LIRMM), University of Leeds, Chapel Allerton Hospital, Leeds LS7 4SA, UK; h.pandit@leeds.ac.uk; 5Department of Oral Pathology, Faculty of Dentistry, Suez Canal University, Ismailia 3, Ismailia Governorate 8366004, Egypt

**Keywords:** tissue engineering, biotribology, cartilage lubrication, osteoarthritis treatment, intra-articular injections

## Abstract

Cartilage degeneration is a characteristic of osteoarthritis (OA), which is often observed in aging populations. This degeneration is due to the breakdown of articular cartilage (AC) mechanical and tribological properties primarily attributed to lubrication failure. Understanding the reasons behind these failures and identifying potential solutions could have significant economic and societal implications, ultimately enhancing quality of life. This review provides an overview of developments in the field of AC, focusing on its mechanical and tribological properties. The emphasis is on the role of lubrication in degraded AC, offering insights into its structure and function relationship. Further, it explores the fundamental connection between AC mechano-tribological properties and the advancement of its degradation and puts forth recommendations for strategies to boost its lubrication efficiency.

## 1. Introduction

The prevalence of OA is steadily increasing, particularly among Europe’s aging population, which is estimated to be around 40 million [1,2]. This upward trend is even more pronounced in the United Kingdom, where OA affects 40.5% of every 1000 people annually [3]. In 2011, the surge in arthritis cases led to a significant increase of 165,000 joint replacement surgeries, with OA patients constituting a remarkable 93% of these procedures [4]. Consequently, this increase in surgeries exerts a substantial economic burden, and it is anticipated to rise 7% by 2035 due to various factors, such as age, gender, and body mass index [5,6]. There is a number of mechanical factors that contribute to the development of OA, including work-related activities, participation in sports, musculoskeletal injuries, and obesity [7,8]. To tackle this growing issue and conserve economic resources, a deeper understanding of the AC structure should be achieved, which will help to develop a cost-effective solution to delay the need for total joint replacements.

Synovial joints exhibit a high level of incongruity at the central surface compared to the peripheral surface, resulting in a limited contact area at the interface. This lack of congruency makes these joints susceptible to surface damage when subjected to high loads [9,10]. The primary components determining the frictional and mechanical properties of AC are mainly collagens, particularly type 2 collagens (55–75%), proteoglycans (15–30%), and lipids, specifically phospholipids (10%). These components vary across the superficial (10%), middle (40–60%), and deep (30%) zones of the AC. Collagen fibers are densest in the superficial zone (1–200 µm thickness, 5–20 nm diameter), providing resistance against compressive, tensile, and shear forces. In the middle and deep zones, collagen fibers have a similar size, with a diameter of 30–5000 nm. Proteoglycan density is lower in the superficial zone but increases in the middle and deep zones. The arrangement of extracellular matrix consisting of collagen fibers and proteoglycans in each zone contributes to the mechanical strength of the AC, with parallel arrangement in the superficial zone and more random arrangements in the middle and radial arrangement in the deep zones [11,12,13,14,15,16,17]. The presence of AC layers covering synovial joints acts as a protective barrier, enabling smoother movements, as shown in Figure 1. These layers facilitate near-frictionless motion, with friction coefficients ranging from 0.005 to 0.025 due to their specific structural and lubricating characteristics [18]. The AC structure exhibits both viscoelastic and poroelastic material properties and typically measures 1–4 mm in thickness in a healthy joint [19]. The elasticity modulus of the AC surface ranges from 0.5 to 10 MPa, which helps with its high load-bearing ability [20]. Considering this complex structure of AC and its role as a protective layer to the synovial joints, their metabolic activities are influenced by various factors, such as matrix composition, soluble mediators from synovial fluid (SF), mechanical load, and pressure [21,22].

The junction between synovial joints secretes SF, which plays a crucial role in joint function. SF acts as both a shock absorber by providing cushioning and as a natural lubricant by facilitating smooth and near-frictionless movement at the joints [23,24]. It is composed of several major constituents, including hyaluronic acid, albumin, lubricin, and globulin, which contribute to its lubrication mechanisms [25,26]. Each of these constituents has a specific role in enhancing lubrication properties. At low shear rates, hyaluronic acid acts as a viscosity modifier for SF by influencing its flow characteristics. Albumin helps protect the AC from wear and maintains its integrity [27,28]. Lubricin plays a role in reducing the shear strength of the contacts between asperities at the synovial interface, which contributes to smoother movement [29,30]. Lastly, globulin contributes to boundary lubrication at the joint interface, which further reduces friction. Collectively, the composition of SF and the specific functions of its constituents contributes to the lubrication mechanisms that allow for optimal joint functionality and minimize wear and friction within the synovial joints.

Despite the well-designed structure of the AC and the presence of effective natural lubrication provided by SF, abnormalities can occur in synovial joints, causing AC degradation and leading to tissue remodeling, as shown in Figure 2. The damage in the OA cartilage is shown using the synovial inflammation, cartilage degradation, bone remodeling, osteophyte formation, and neurogenesis [31]. OA also leads to the thickening of the calcified zones that separate the AC from the underlying calcified cartilage. This calcification of the AC is accompanied by the presence of calcium pyrophosphate dehydration [32,33]. Moreover, OA causes synovial infiltration, characterized by the presence of low-grade macrophages and lymphocytes in the early stages of the disease, leading to synovitis (inflammation of the synovium) [34]. These abnormalities are driven by inflammatory mediators present in the joints that result in OA [35,36]. In OA, there is an increase in cell proliferation, leading to the formation of cell clusters and an upregulation of matrix proteins and matrix-degrading enzymes. This disruption of chondrocytes is considered an injury response, triggering matrix remodeling, cartilage calcification, and abnormal hypertrophy-like maturation within the cartilage [37,38]. A variety of techniques have been utilized to detect structural abnormalities resulting from OA. These include computed tomography, X-ray imaging, ultrasonography, and physical examination. Additionally, alternative methods like vibroarthrography Raman spectroscopy, Fourier transform infrared scanning, etc., have been explored for this purpose [39,40,41]. Based on these methodologies and the extent of damage in OA cartilage, it is commonly classified according to the Kellgren–Lawrence (K-L) system, which includes grades 0 to 4. Grade 0 indicates a healthy AC, while grade 4 indicates a severely damaged AC [42].

This review aims to provide a comprehensive overview of the advancements made in understanding the lubrication mechanisms of AC. By conducting mechano-tribological tests, researchers have gained insights into the degradation of AC, particularly in cases of OA. These tests have revealed changes in the properties, surface characteristics, and structural integrity of AC, shedding light on the underlying lubrication mechanisms. Recent developments in tribological testing methodologies, such as stationary, migrating, and convergent stationary contact area, have significantly contributed to our understanding of cartilage tribology. These testing setups have also provided new insights into theories regarding the progression of OA in AC. Building on a brief background on AC, each section of this review deals with a specific AC lubrication mechanism, covering research summaries, testing methods, quantitative explanations of AC properties, and their relationship to AC degradation. Finally, the report concludes with an understanding of AC degeneration in the case of OA based on the failure of the lubrication mechanism at the joint interface. Finally, the importance of temporary solutions using intra-articular injections is summarized.

## 2. Mechanical Properties of AC in OA

AC serves as a protective covering for bones and exhibits different biomechanical properties across synovial joints. Its elastic nature allows it to deform under load, with collagen fibrils, proteoglycans, and interstitial water being key contributors to these properties. Collagen fibrils provide tensile strength, while proteoglycans and interstitial water contribute to compressive strength. Experiments such as confined, unconfined, and indentation tests have been conducted to assess aggregate modulus, permeability, compressive stiffness, shear strength, and tensile strength across various specimens, as tabulated in Table 1. In healthy AC, the aggregate modulus and permeability are calculated from the creep behavior of the cartilage compression, where the load is constant, allowing for the displacement of the cartilage. This aggregate modulus measurement is of the stiffness of the tissue at equilibrium; further, this stiffness is the result of a change in glycosaminoglycan and water content. Similarly, the tensile stiffness of the AC is due to the collagen content and the permeability is due to interstitial water variation at the surface interface.

In cases of OA, the cartilage surface wears out at heavily loaded regions like the medial area of the femur and tibia. This results in a loss of the superficial lining, which increases surface permeability, allowing interstitial fluid to move in and out rapidly (Figure 3). Deformation-dependent permeability is the property of AC that helps with load sharing between the solid and fluid composition in a healthy condition. However, having rapid fluid flow within the cartilage matrix due to increased permeability in the case of OA causes uneven distribution of the load, causing high stress on the solid matrix [11]. Based on these mechanical tests, researchers have highlighted the differences between healthy and degraded cartilage across species. Lower modulus values and increased permeability have been found to be the characteristics of degraded cartilage [43]. For example, bovine samples exhibit decreased aggregate modulus in degraded cartilage (from 0.37 MPa to 0.06 MPa in confined compression and from 3.9 MPa to 0.7 MPa in indentation tests). Similarly, porcine samples show reduced modulus values for degraded cartilage (from 0.71 MPa to 0.09 MPa under confined compression). Human OA cartilage samples follow a similar trend, with modulus values decreasing to a minimum of 0.07 MPa, indicating lower mechanical strength. Additionally, the unique morphology of human knee joints is influenced by bipedal locomotion, which results in increased compressive forces at the medial condyle compared to the lateral side. Consequently, this increases stress in the medial region, which progresses the cartilage degradation [44,45].

**Table 1 bioengineering-11-00541-t001:** An overview of the mechanical testing conducted on AC across various species, including details regarding the experimental conditions, testing setup, and quantitative data obtained from these experiments.

Sample Source	Reference	Healthy/Osteoarthritic	Natural OA/Induced OA	Cartilage Region	Testing Condition	Mechanical Properties		
						Young’s Modulus	Aggregate Modulus	Other Findings
Bovine	[46]	H	-	Knee joint	Static and dynamic confined compression (microscale)	-	0.37 ± 0.03 MPa (adult) 0.43 ± 0.02 MPa (calf), and 0.15 ± 0.01 MPa (fetus)	Permeability (k_p_) expressed as (log_10_k_p_(m^2^/(Pa s)) −14.92 ± 0.93 (adult) −15.19 ± 0.32 (calf), and −15.60 ± 0.46 (fetus)
	[47]	H	-	Knee joint	Unconfined compression (macroscale)	14.6 ± 6.9 MPa at 0.1 Hz to 28.7 ± 7.8 MPa at 40 Hz	0.49 ± 0.10 MPa	Peak compressive strain amplitudes 15.8 ± 3.4% at 0.1 Hz to 8.7 ± 1.8% at 40 Hz
	[48]	H	-	Knee joint	Unconfined compression (microscale)	-	0.96 ± 0.47 MPa (adult) 0.89 ± 0.39 MPa (calf), and 0.72 ± 0.36 MPa (fetus)	Poisson’s ratio 0.26 ± 0.11 (adult) 0.09 ± 0.02 (calf), and 0.11 ± 0.03 (fetus)
	[49]	H	-	Knee joint	Indentation (microscale)	3.9 ± 0.7 MPa (Effective contact modulus)	0.62 ± 0.10 MPa (equilibrium contact modulus)	Tensile modulus 4.3 ± 0.7 MPa and permeability 2.8 ± 0.9 × 10^−3^ mm^4^/Ns
	[50]	H	-	Knee joint	Indentation (microscale)	-	0.93 MPa (equilibrium contact modulus)	-
	[51]	OA	(In vitro) induced with type II bacterial collagenase	Knee joint	Confined compression (macroscale)	-	0.06 ± 0.03–0.13 ± 0.06 MPa	Permeability 4.73 ± 1.43 × 10^−14^ m^4^/N s–8.25 ± 2.24 × 10^−14^ m^4^/N s
	[52]	OA	(In vitro) induced using collagenase, chondroitinase ABC, or elastase	Knee joint	Indentation (microscale)	-	0.7 MPa (collagenase), 0.3 MPa (chondroitinase ABC), and 0.7 MPa (elastase)	-
	[53]	OA	(In vitro) induced using collagenase	Knee joint	Unconfined compression (microscale)	-	0.45 ± 0.21 to 0.23 ± 0.14 MPa with 2 U/mL collagenase treatment and 0.49 ± 0.19 to 0.19 ± 0.08 MPa with 10 U/mL collagenase treatment	Compressive strain 21.7 ± 5.6 to 26.2 ± 7.6% at 0.1 Hz loading frequency and from 9.6 ± 3.3 to 13.5 ± 3.2% at 40 Hz loading frequency with 10 U/mL collagenase treatment
Porcine	[54]	H	-	Knee joint	Indentation (microscale)	2 MPa at 2.5 mN and 7 MPa at 10 mN	-	Contact stiffness 0.5 kNm^−1^ at 2.5 mN and 4.0 kNm^−1^ at 10 mN Hardness 0.07 ± 0.01 MPa at 2.5 mN
	[55]	H	-	Knee joint	Indentation (mesoscale)	2.93 MPa	-	Hardness 0.05 MPa
	[56]	H	-	Knee joint	Confined compression (micro scale)	-	0.71 ± 0.50 MPa (creep) and 0.68 ± 0.48 MPa (recovery)	-
	[57]	OA	(In vitro) induced with papain	Knee joint	Confined compression (microscale)	-	0.09–0.38 MPa (medial femoral condyle), 0.32–0.42 MPa (lateral patellar groove), and 0.095–0.38 MPa (medial patellar groove)	(1.9–7) × 10^−15^ m^4^/N s (medial femoral condyle), (1.2–2.6) × 10^−15^ m^4^/N s (lateral patellar groove), and (1.2–1.5) × 10^−15^ m^4^/N s (medial patellar groove)
Rabbit	[58]	H	-	Knee joint	AFM indentation (nanoscale)	-	0.52 ± 0.05 MPa (superficial zone) 1.69 ± 0.12 MPa (calcified zone)	Surface roughness 59.0 ± 12.6 nm
	[59]	OA	(In vivo) intramuscular injection of ketamine (100 mg/kg) and xylazine (8 mg/kg)	Knee joint	Surface properties	-	-	Surface roughness values (mean rms values) 95–320%
	[60]	OA	(In vivo) anterior cruciate ligament transection (ACLT) model	Knee joint	Indentation (nanoscale)	3.37 ± 1.23 MPa (instantaneous modulus)	0.85 ± 0.29 MPa (equilibrium modulus)	-
Human	[61]	H	-	Knee joint	Confined compression (macroscale)	-	0.499 ± 0.208 MPa to 1.597 ± 0.455 MPa)	Permeability 0.689 ± 0.304 × 10^3^ (mm^4^/N-s) to 1.318 ± 0.673 × 10^3^ (mm^4^/N-s)
	[62]	H	-	Knee joint	Unconfined compression (macroscale)	-	1.60 ± 0.51 MPa to 2.47 ± 0.49 MPa	-
	[63]	H	-	Knee joint	Unconfined compression (microscale)	-	0.53 ± 0.25 MPa	-
	[64]	OA	Total joint replacement patients	Knee joint	Unconfined compression (macroscale)	-	-	Shear modulus 4.6 ± 1.8 MPa
	[65]	OA	Total joint replacement patients	Knee joint	Indentation (macroscale)	2.51 to 10.7 MPa (instantaneous modulus)	0.07 to 2.86 MPa (equilibrium modulus)	-
	[66]	OA	Total joint replacement patients	Knee joint	Micropipette aspiration technique	-	Chondrocytes (0.63 ± 0.51 kPa), instantaneous modulus, and 0.33 ± 0.23 kPa) equilibrium modulus	-

(H—healthy, OA—osteoarthritis, AFM—atomic force microscopy).

## 3. Tribological Properties of AC in OA

Bio-tribology explores the surface interaction of AC in synovial joints using both biological and modeled samples [67,68]. Various experimental setups have been designed to investigate the surface interaction of AC, which include pin-on-plate configurations (such as cartilage on a glass plate or a cartilage pin on a cartilage plate), pin-on-disk setups (using a spherical probe made of metal or glass on cartilage discs), and pendulum friction simulators [67,69,70]. Researchers have also mimicked the natural properties of AC by examining quantitative properties such as surface roughness, coefficient of friction, and wear rate. However, replicating the tribological properties and engineering a tissue that mimics the physiological conditions of AC pose significant challenges [71,72]. Traditional tribology tests have estimated the friction coefficient of AC to range from 0.005 to 0.025 [73]. These experimental setups deal mainly with understanding the tribological behaviors of the cartilage in different physiological conditions. The three important types of contacts used to study cartilage tribology are stationary (SCA), migratory (MCA), and convergent stationary contact area (cSCA) setups. Ateshian and Wang first observed this phenomenon while sliding a glass sphere probe onto the AC surface, referring to it as the MCA, which resulted in lower friction coefficients [74]. Conversely, when the cartilage was paired with a flat glass surface, the friction coefficients increased, leading to what they called the SCA [75]. These two contact configurations, the MCA and SCA, are utilized to study cartilage tribology to investigate interstitial fluid pressurization and migration at the AC contact interfaces. The c-SCA configuration is the modified version of SCA configuration that was developed solely for experimental purposes to explain the concept of elasto-hydrodynamic lubrication in larger cartilage plugs. All three setups are shown in Figure 4 [76].

Researchers have utilized the developed experimental setups to investigate various tribological properties of AC. These experiments have calculated COFs for AC across different species, as summarized in Table 2. In the case of animal cartilage, the COF values for healthy bovine cartilage range from 0.024 to 0.2, depending on the parameters used. For healthy porcine cartilage, COF values range from 0.001 to 0.14, while healthy human cartilage exhibits a COF value of 0.22. In contrast, degraded cartilages show increased COF values for all samples. For degraded bovine samples, COF ranges from 0.17 to 0.19, and in the case of human samples, COF values gradually increase based on the grades of OA up to 0.409. The observed changes in the trends of frictional behavior raise a crucial question: Does the tribological alteration on the surface of the cartilage contribute to the development of OA, or does OA lead to changes in tribological behavior? This is a significant question that this article aims to address by the end.

## 4. Chronology of Cartilage Lubrication

Numerous lubrication models proposed to explain the characteristics of AC in synovial joints aiming to minimize friction and wear are shown in Figure 5. These models consider different loading and motion conditions experienced by the joints. Knowledge of joint lubrication dates back to the medieval period, when Paracelsus introduced the term “synovia” in the sixth century [89]. In 1691, Harvey studied the lubrication properties of SF, which contributes to smooth joint movements [90]. In recent years, extensive research, both theoretical and experimental, has been conducted to understand the reasons behind the low friction in synovial joints.

In 1932, MacConaill proposed the use of hydrodynamic lubrication theory considering the wedge-like fluid film formation between the AC surfaces [91], which was later supported by Jones’ friction measurements of synovial joints in 1936 [92]. However, this theory did not consider the incongruent nature and low-speed motions of the joints, which are unfavorable for creating hydrodynamic lift to form the wedge. In 1959, McCutchen explained that AC has porous walls and that when the load is applied, the fluid is pushed out to the interface and supports the load. This weeping nature of porous cartilage was termed “weeping lubrication” [93]. In 1963, Dintenfass modified this theory by considering the deformability of the AC structure and the viscous resistance of SF, calling it “elastohydrodynamic lubrication” [94]. Hooke and O’Donoghue further studied this modification; however, the calculated film thickness in this lubrication model was smaller than the surface roughness of the AC surfaces at the contact interface [95]. To address this issue, Dowson and Jin proposed the concept of micro-elastohydrodynamic lubrication, taking into account that AC surfaces have flat asperities under physiological pressure [96]. Yet, this theory could not explain the time-dependent variation in frictional properties of AC. Following studies on fluid film lubrications, Mow and Lai proposed a self-generating lubrication mechanism considering the biphasic nature of AC [97]. They suggested that fluid exuded at the edges of the tissue is reabsorbed in the center, contributing to lubrication and reducing friction at the AC surface interface. This model provided visual evidence for the flow behavior of AC using an optical sliding contact rheometer. The high load-bearing capacity of synovial joints was explained by the squeeze film lubrication theory [98]. Fein studied this in 1967 [99], and Higginson and Unsworth later demonstrated it [100,101]. They explained that the AC’s ability to bear high loads is due to the time-varying pressure field created by the lubricant’s viscous resistance as it is squeezed from the gap. During squeeze film conditions, water content from SF passes into AC over the contact region, while the remaining solute content, such as hyaluronic acid protein, acts as a lubricant. This theory was termed “boosted lubrication” and was later supported by qualitative and quantitative characterizations [102]. In 1989, based on the boundary conditions of the AC–SF interface, Hou and colleagues formulated the biphasic squeeze film model [103]. This inspired Hlavacek in 1993 to develop the first biphasic model for SF, combining the boundary lubricant nature of solute composition in SF as viscous and non-Newtonian with Mow’s biphasic theory of cartilage [104]. This theory explained that the gel formed during squeeze film due to hyaluronic acid is minimal and contributes very little to boosted lubrication (≤1 s). Researchers have also studied boundary lubrication models by varying the lubricant contents in water to evaluate their effect on the friction coefficient. While these studies have shown improvements in the friction of AC, these improvements alone are not significant enough to fully explain friction reduction or joint lubrication.

Considering the complexities of the synovial joint mechanisms, it is not possible to explain the physical aspects of joints with one individual lubrication regime. Hence, a combination of different lubrication regimes explains different gait cycles, giving rise to mixed or multimode lubrication regimes, as shown in Figure 6. The aspects of speed and eccentricity in the joint movements are not enough to maintain the fluid film in contact, which creates the solid–solid interaction and gives rise to the mixed regime [105]. Further, researchers have conducted experiments to study interstitial fluid pressurization in the joint interface and to understand the role of the solid matrix, which represents the collagen proteoglycan network, along with the fluid phase, representing the interstitial water with dissolved ion phases [74,106,107,108]. The interactions between these phases and their contribution to friction reduction on the AC surface have been explained using biphasic and triphasic theories developed by Mow, Lai, and their colleagues [109,110]. These theories provide an explanation for the pressurization and flow of interstitial water into the porous and permeable solid matrix when the joint is loaded. Experimental results have confirmed these theories by describing compression creep, stress relaxation, and dynamic relaxation of AC. Investigations have also been carried out on these theories to understand cartilage rehydration under conditions that do not involve unloading or induced migration of AC [74,111]. Further studies on tribological rehydration explains the fluid retention and recovery of joint conditions [78,112]. The study of AC lubrication and wear has been conducted using various theories, often with contradictory results, as summarized in Table 3 [113]. However, it is crucial to understand the fundamental mechanisms of cartilage lubrication to develop therapeutic solutions for joint degeneration such as OA. The primary focus of this review is to comprehend the AC lubrication models mentioned above to gain insights into the progression of OA.

### 4.1. How Lubrication Models Fail in the Case of OA

Previous discussions regarding lubrication models for AC can be categorized by drawing from established tribological lubrication models, which include fluid film, mixed, and boundary lubrication. These models have been adapted to account for the structure and function of AC, resulting in modifications to these lubrication models. To examine the concept of fluid film formation within AC, several models have been developed, including hydrodynamic, weeping, elastohydrodynamic, micro-elastohydrodynamic, and tribological rehydration models. Additionally, models have been proposed based on the formation of mixed film modes in AC, such as osmotic, squeeze film, boosted, biphasic, and triphasic models. Lastly, a hydration model has been explained considering the lubrication effects of synovial constituents and their boundary lubrication properties. All these models focus on comprehending how healthy cartilage is lubricated with SF at the interface to ensure smooth functioning. However, when it comes to OA, the effectiveness of these models in explaining the progression of cartilage degeneration becomes questionable. In this section, we will dive into each model to assess its relevance in the context of OA.

#### 4.1.1. Fluid Film Lubrication

Fluid film lubrication theory was originally developed within the conventional context of tribological lubrication in journal bearings. The interface is separated by the pressure generated by the lubricant film at the contact point, which has been adapted for understanding AC lubrication in synovial joints. This adaptation has led to the emergence of various theories, including hydrodynamic, hydrostatic (or weeping), elastohydrodynamic, micro-elastohydrodynamic, and tribological rehydration models. These lubrication models have been formulated while considering the function and physiological characteristics of AC.

The proposal of a hydrodynamic lubrication mechanism is based on the notion that SF in synovial joints creates a wedge-shaped fluid layer at the articulating junction, as shown in Figure 7a [91,116]. This fluid layer prevents direct contact between the two surfaces by acting as a thin lubricant and generating a hydrodynamic lift (about 10 µm [136]). However, unlike typical hydrodynamic bearings that require high-speed transverse motion to establish this lift (500 to 4000 rpm for 40 mm-radius bearings [137]), the pin-on-plate experiments performed by Gleghorn et al. using bovine AC failed to obtain the hydrodynamic lubrication even at a high speed of 50 mm/s and a low strain of 5% [77]. The non-conformal nature of biological joint surfaces and the absence of consistent high-speed and lighter loads make the contacting geometries unfavorable for this mechanism. Also, research has highlighted the critical role of a healthy meniscus in maintaining optimal fluid load support within the knee’s AC. The low permeability of a healthy meniscus serves to limit fluid exudation from the AC, thereby aiding in joint protection against OA [138]. Thus, it is evident that meniscus damage in OA leads to lower fluid load support by the AC. The properties of SF in OA conditions can vary due to aging or damage at the articulating surfaces. The reduction in constituents such as hyaluronic acids and lubricin in diseased joints can affect the viscosity of the SF, which may not support interstitial fluid pressurization (viscosity was reduced by a factor of 20 at a shear rate of 10^−1^ s^−1^ and by a factor of 3 at a shear rate of 10^3^ s^−1^) [139,140,141]. Hence, hydrodynamic lubrication is insufficient to explain the degradation mechanism of OA.

Hydrostatic/weeping lubrication was introduced by McCutchen in 1959, drawing inspiration from hydrostatic bearings [117,118]. According to this theory, when two articulating surfaces are compressed, the interstitial fluid is forced out of the AC pores, resulting in a pressurized fluid film that supports the solid matrix, as shown in Figure 7b [115,142]. Consequently, the frictional forces primarily act on the portion of the load that is transferred across the solid matrix, which signifies the load-sharing mechanism [93]. These findings were substantiated through experimental observations of cartilage on glass over a period. Further, they demonstrated that the frictional force did not decrease when the load was removed for a brief period of 1 s but rather increased upon reapplication of the load [142]. This mechanism provided an explanation for the time-dependent frictional behavior of AC, which could not be accounted for by hydrodynamic or boundary lubrication theories. However, this proposed mechanism sparked controversies regarding the principle of conservation of linear momentum [143]. The concept of hydrostatic lubrication cannot be applied in the case of OA due to the changes in the permeability of the cartilage surface. Researchers have identified that the permeability of the cartilage changes in the extracellular matrix structure allow for rapid fluid flow and cause uneven deformation [11]. This permeability is elevated from 4.19 ± 3.78 × 10^−17^ m^4^/N s to 10.2 ± 9.38 × 10^−17^ m^4^/N s, influencing the biomechanical properties of the chondrocytes in the AC tissue [144]. Thus, the fluid load support, as mentioned in the weeping model, fails to lubricate the degraded cartilage interface.

Previous lubrication models did not take into account the deformation of AC as a factor in explaining the frictionless motion of synovial joints [94]. Dintenfass developed an AC lubrication model based on elastohydrodynamic theory, considering the viscous resistance of the SF (viscosity of SF is 10 Ns/m^2^ at 0.1 s) and the elastic deformation (10.78 MPa when compressed in water) of the AC surface, as shown in Figure 7c [145]. This model was supported by the observation that the viscosity of SF is negligible under pressure and that the elasticity modulus of the AC is very small [145,146]. Therefore, when a high load is applied, the film thickness formed at the interface is minimal due to the flattening of the AC surface and the increased contact zone. This flattening phenomenon of the AC surface under physiological load was later referred to as the micro-elastohydrodynamic effect [96,147]. However, for the elastohydrodynamic lubrication, the film thickness at the AC interface for an healthy joint was in the range of 0.1 to 1 µm, with a surface roughness of 2–5 µm; thus, this theory was substantiated by various theoretical calculations and experimental results [148]. Tanner also calculated the film thickness of hip joints during normal walking speeds, yielding a thickness of 10^−5^ cm or greater at a 20 cm^2^ area with a friction coefficient of 0.003 [136]. These experimental results demonstrated the formation of a film thickness during relative motion. But in terms of the degraded AC, the surface roughness increases due to superficial fronding at the upper layer. The surface roughness with arithmetic average absolute values, the maximum peak heights, and the mean spacing between local peaks of the OA cartilage surface were 71%, 80%, and 51%, respectively, compared to the healthy cartilage [149]. These studies show that the load-bearing ability of the AC cannot be explained using the micro-elastohydrodynamic or elastohydrodynamic lubrication models.

#### 4.1.2. Boundary Lubrication

When the hydrodynamic lubrication mechanism was proposed, it faced skepticism from researchers for not considering the contribution of synovial compositions. In 1959, Charnley postulated that the presence of boundary lubrication on the AC surface was due to the adherence of the SF at the AC interface [124,150,151]. The adherence, or boundary action, is governed by the glycoprotein fraction of SF, which forms a surface film with a thickness ranging from 1 to 100 nm [152]. Swann et al. demonstrated that polypeptide chains within SF function as boundary lubricants when isolated from the SF [125,126]. Subsequently, researchers began investigating the individual components of SF to determine their contributions to the nature of the boundary lubrication [127]. The main constituents identified were hyaluronic acid, aggrecans, lubricin, and phospholipids. Several reviews have outlined the significant contributions of these constituents and the techniques employed to study their boundary lubricant properties [153,154]. The complete explanation of the boundary lubrication mechanism at the articulating surfaces of biological joints cannot be solely attributed to the presence of hyaluronic acids, lubricin, and phospholipids constituted in SF [155]. Through in-depth studies on friction conducted using surface force balance measurements, it has been observed that hydrated ions are trapped within the phospholipid solutions [156]. These hydrated ions act as effective lubricants by creating a strong repulsion during compression, facilitated by the repulsive hydration shells surrounding the trapped counterions. This phenomenon is referred to as the hydration lubrication mechanism (Figure 8) [128,157]. The underlying reason for this low-friction behavior is the formation of a slip plane between closely packed phosphocholine liposomes, wherein the outer surfaces of liposomes oppose each other within a group [158]. Studies on degraded cartilage indicate that an increase in surface roughness and permeability of the cartilage leads to an extension of the boundary-mode plateau, resulting in higher friction at the interface. This is primarily attributed to the loss of boundary lubricants present in the SF at the AC interface. To counteract this, boundary modes can be achieved by either increasing the viscosity of the lubricants (for instance, if the lubricant viscosity for healthy cartilage is 150 mPas, degraded cartilage might require 1500 mPas to achieve healthy friction values) or by elevating the sliding speeds at the interface [159]. By understanding and harnessing this mechanism, it becomes possible to develop intra-articular injections containing suitable lubricants that effectively reduce friction. Furthermore, this knowledge can be applied in the field of tissue engineering scaffolds for diseased cartilage [155].

#### 4.1.3. Mixed-Mode Lubrication

The film thickness formation in the case of hydrodynamic lubrication and the boundary effect of the synovial constituents cannot completely explain the complex lubrication mechanism of the synovial joints. There must be different modes of lubrication that comes together in this complex structure that help explain its low friction and wear properties [106]. This mixed lubrication regime can be called adaptive multi-mode lubrication, which can explain each individual phase of gait cycles. Based on this knowledge, various lubrication models, such as osmotic, squeeze film, boosted, biphasic, and triphasic lubrication models, have been proposed.

The squeeze film lubrication model proposed by Dowson in 1966 suggests that during joint motion the SF between the AC surfaces in synovial joints experiences compression (around 18 MPa under the physiological loading condition [160]), resulting in the generation of pressure [161]. This pressure is attributed to the resistance offered by the lubricant as it is squeezed out by the contact surfaces, as shown in Figure 9a. Consequently, the articulating surfaces are separated by a fluid film, leading to a localized depression where the lubricant becomes trapped in supporting that load [162]. Similarly, Maroudas and Walker et al. proposed boosted lubrication, which suggests that when two articulating surfaces are compressed, the solvent component of SF enters the pores of the AC surface, as shown in Figure 9b. The solute or concentrated part of SF remains behind, providing lubrication, and this phenomenon is called “boosted lubrication” [102,163,164,165,166]. Both squeeze film and boosted lubrication fails in the case of OA due to the loss of proteoglycans, which leads to a decrease in the dynamic and equilibrium compressive modulus of the AC [167,168]. This reduction allows for increased hydraulic permeability, resulting in decreased interstitial fluid load support during loading, ultimately leading to more cartilage damage [167].

Previous lubrication mechanisms were unable to provide a satisfactory explanation for the time-dependent frictional behavior exhibited by cartilage [82]. This behavior can be understood by examining the stress relaxation or creep nature of AC, which are directly associated with the release of interstitial fluid from the surface during tissue creep [75,131,132,169,170,171]. These properties of AC were elucidated by the biphasic theory proposed by Mow et al. When AC is subjected to compression or tension, the flow-dependent and flow-independent behaviors, as well as the mechanism of viscoelasticity, can be understood based on the change in the modulus values and the stress relaxation experiments (Figure 9c) [134,172]. This theory provides an explanation for the viscoelastic behavior of AC in various configurations, including tension, compression, and indentation [173]. According to the biphasic theory, AC is composed of a binary mixture consisting of an incompressible elastic solid (collagen and extracellular matrix) and an inviscid incompressible fluid (interstitial fluid) [174,175]. Various researchers reported experimental measurements of interstitial fluid pressures calculating the aggregate elastic modulus of AC from creep experiments to be around 0.7 ± 0.09 MPa and from stress relaxation experiments to be around 0.76 ± 0.03 MPa [174,175,176]. Oloyede and Broom demonstrated this phenomenon through confined compression creep-loading ex situ experiments [177], while Soltz and Ateshian explained it using AC behaviors such as creep, stress relaxation, and dynamic loading in both confined compression (aggregate compression modulus = 0.64 ± 0.22 MPa and axial permeability = 3.62 ± 0.97 × 10^−6^ m^4^/Ns) and unconfined compression ex situ experimental setups (tensile modulus = 12.75 ± 1.56 MPa and radial permeability = 6.06 ± 2.10 × 10^−16^ m^4^/Ns ) [133,175]. These findings have established that the viscoelastic behavior of AC is solely influenced by intrinsic mechanical properties [178] such as porosity and permeability of the solid phase in AC, as well as the frictional resistance arising from the drag exerted by the interstitial fluid. Furthermore, a state of stress relaxation equilibrium can be achieved when the flow of interstitial fluid comes to a halt. In this equilibrium state, the entire load is borne by the solid matrix, providing an explanation for the load deformation response observed in AC tissue [179,180].

As an extension of biphasic theory, triphasic theory was put forth, which introduces the concept of an additional phase known as the ionic phase, which is distinct from the interstitial fluid. This model examines the mechanical properties of AC by incorporating three distinct phases: the solid phase, the interstitial fluid phase, and the ionic phase. In this model, the solid phase of the tissue is described as a homogeneous, isotropic, linearly elastic material that experiences infinitesimal strain within the interstitial fluid phase [134,135,175]. Considering these models, effective lubricant-based solutions can be identified depending on the properties of the AC. However, in the case of degraded cartilage, the alterations, including elevated water content within the AC structure, hinders the generation of proper fluid pressure during applied loading. Furthermore, the degenerative process impacts the hydraulic permeability of the superficial zones, thereby influencing the flow-dependent and flow-independent viscoelastic behavior of AC.

### 4.2. Concept of Tribological Rehydration

Previous studies have extensively examined various fundamental concepts of AC to elucidate its low-friction characteristics. These investigations have highlighted the significant role of interstitial fluid within the cartilage in facilitating both its load-bearing capacity and its smooth, frictionless motion [181,182,183]. Several studies, such as those on weeping [142], boosted [102], biphasic [175], and hydration lubrication [157], have primarily focused on interstitial fluid pressurization and its influence on joint movement. However, most of these studies have examined the process of fluid exudation through controlled experiments involving static loading [29,184,185]. In contrast, the mechanisms of interstitial rehydration in AC remain unclear; however, the concept of hydrostatic/weeping lubrication by McCutchen [117,118] covered the time-dependent interstitial fluid movement. Also, he later proposed an explanation for interstitial rehydration as the osmotic pressure gradient resulting from the density of fixed charges in the tissue, which creates a driving force for AC rehydration [78,122,123,186,187]. There is only a limited number of studies that have specifically addressed this aspect. Nevertheless, during joint articulation, there is evidence of tissue thickness recovery, suggesting the presence of an additional mode of tissue recovery [78]. Consequently, various investigations have been conducted to comprehend motion-induced tissue recovery, ultimately revealing the migratory nature of joints [187]. This concept has led to the identification of a novel phenomenon known as tribological rehydration, wherein the AC can regain interstitial hydration while sliding within a continuously loaded stationary contact area, thereby contributing to contact migration [78,123]. However, the degraded OA cartilage has uneven surfaces across the contact regions that could possibly avoid contact migration at the loaded regions.

## 5. Relevance of AC Lubrication Theories to OA

Comprehensively addressing cartilage degeneration, particularly OA, necessitates detailed knowledge of the lubrication process within the structure of AC. As previously mentioned, the frictional properties of the AC primarily depend on its lubrication. However, due to pathological changes occurring in the synovial joints, the AC surface becomes softer and more permeable, thereby affecting its inherent self-lubricating mechanisms [188,189,190]. These changes are a result of alterations in joint mechanics and the aging process. Initially, these changes disrupt the mechanisms responsible for joint lubrication, leading to joint pain [37,38]. Over time, if left untreated, they progress and eventually result in the complete breakdown of the AC, leading to severe joint dysfunction and the need for surgery [191,192,193]. Mechanical stresses are commonly observed as precursors to this degenerative process, causing surface alterations such as microcracks, microcraters, and peeling of the superficial layer of the AC [194,195]. Additionally, it has been suggested that the penetration of SF enzymes through the AC surfaces can lead to the breakdown of the matrix ground substance, resulting in tissue softening and contributing to OA [196,197]. All these factors collectively disrupt the normal lubrication mechanisms within the joints and interfere with the nutritional pathways necessary for maintaining the health of the AC. It is evident from the discussion that both the mechanical and physiological functions of the AC are vital in providing proper lubrication mechanisms. Unfortunately, these factors are often compromised, thereby compromising the self-lubricating nature of the AC.

The application of classical lubrication theories to synovial joints reveals that no single lubrication theory can fully account for the overall tribological behavior of these joints. This underscores the complexity of the biological nature of synovial joints, particularly in their response to varying velocities, loads, and motions. From this comprehensive understanding, it becomes evident that synovial joints do not rely on a single lubrication mechanism but rather operate through a combination of multiple modes. However, in the case of OA, where the AC surface is damaged, its normal functioning is compromised, often leading to a reliance on one or two specific lubrication mechanisms due to physical constraints. Friction at the interface of AC is primarily influenced by lubrication, which is sensitive to changes in normal and tangential forces as well as relative velocities over time. When subjected to higher loads, the ability of interstitial fluid to provide support decreases, resulting in the solid extracellular matrix bearing most of the compressive and frictional loads [107]. This is when the properties of boundary lubrication in AC become significant, emphasizing solid–solid interactions [198,199]. However, in cases of OA, the initial cartilage damage involves surface irregularities, matrix deterioration, and progressive wear, as seen in the K-L classification. This damage progressively increases from grade 1 to 4 via structural changes, which negatively impacts the lubrication properties. Further, this damages the lubrication by the interstitial fluid, leading to increased reliance on boundary lubrication at the surface. Therefore, understanding the effects of SF constituents like hyaluronic acid, lubricin, and phospholipids on boundary lubrication is crucial for developing therapeutic approaches aimed at enhancing AC lubrication efficiency. This provides information to identify therapeutic approaches that can help restore function during the early stages of these diseases. Numerous therapeutic strategies have been explored to reduce pain, slow down cartilage degradation, and enhance the functioning of AC [200,201,202,203].

### Lubricant-Based Solutions

Lubricant-based injections serve as a temporary remedy for managing AC function in OA conditions. These injections are designed to provide lubrication at the interface and regulate normal cartilage function [204]. However, due to erosion and increased permeability in the superficial layer of degraded AC, many proposed lubrication mechanisms may be ineffective. Nonetheless, lubrication injections still offer temporary relief and can be beneficial for older individuals with OA [205]. SF acts as a natural lubricant at the cartilage interface, containing essential constituents that enhance lubrication. These constituents are used in various lubricant-based injections, facilitating tribosupplementation and reducing friction. Natural lubricants such as hyaluronic acid, chondroitin sulphate, lubricin, and phospholipids possess bifunctional properties, forming loops on the cartilage surface and acting as amphiphilic surfactant molecules [206]. They are modified to form bioinspired molecules, especially hyaluronic acid and lubricin. Furthermore, alternative forms of lubricants are explored for example nanoparticle-based and peptide-based lubricants [207]. In OA conditions, the levels of these constituents significantly decrease compared to healthy samples. Specifically, the concentration of hyaluronic acid declines from 3.12 mg/mL to 0.91 mg/mL, chondroitin sulphate decreases from 18.4 mg/mL to 8.71 mg/mL, lubricin measures at 0.28 mg/mL, and phospholipids measure at 0.2 mg/mL [207]. Furthermore, these natural lubricants serve as inspiration for the development of bioinspired lubricants utilizing long-chain molecules with nanoparticle or peptide-based structures to mimic their characteristics, as shown in Figure 10.

Advancements in lubricant-based injections utilizing bio-inspired lubricants have demonstrated significant improvements in frictional behavior. The summary of these lubricants and their impact on degraded cartilage-enhancing frictional coefficients is presented in Table 4. Several hyaluronic acid-based injections, including Synvisc, Eurflexxa, and Supartz in the United States [208], and Durolane and Ostenil in the United Kingdom [209], have been approved by regulatory authorities. Despite providing initial fluid film lubrication in hyaluronic acid-based and boundary lubrication in lubricin-based injections, as evidenced in the Table 4, these injections exhibit diminished long-term effectiveness due to shear thinning and erosion. The decline in viscosity of these lubricants over time at the cartilage interface post-injection period contributes to their lubrication failure. However, relying solely on a single lubrication mechanism to reduce the friction coefficient has proven to be inadequate. Combining two or three lubrication mechanisms to enhance effectiveness may be a more reasonable approach for achieving long-term efficiency. Ideally, injections should possess compositions that concurrently offer fluid film and boundary effects to prolong their efficacy at the degraded interface. This characteristic is equivalent to natural SF, although SF lacks a sufficient quantity of increased boundary lubricants. Consequently, enhancing the constituents with additional boundary lubricants such as lubricin, long-chain phospholipids, etc., and incorporating these molecules into intra-articular injections can potentially achieve synergistic effects from different lubrication mechanisms. Several studies have been conducted to accomplish this goal, employing approaches such as mesenchymal stem cell-derived therapies and hyaluronic acid gel-enhanced injections in conjunction with osteochondral scaffolds to facilitate early-stage treatments of OA [210,211,212]. Thus, biotribology of AC lubrication emerges as a crucial area for resolving OA issues through improved treatments.

## 6. Conclusions

In conclusion, the study of lubrication in AC offers valuable insights into the knowledge of mechanical and tribological aspects in AC and its degeneration. The failure in the lubrication mechanism of the natural synovial joints is clearly seen as an aging effect or accidental damages resulted due to change in various physical conditions of human body. This change in lubrication mechanism can lead to complete failure of tribological aspects of joints leading to OA. Therefore, the question of whether tribological changes lead to OA can be answered by asserting that these changes in cartilage are the results of OA and are the cause for further progression. Thus, understanding AC lubrication is a potential way to revolutionize OA management, by developing lubricant supplements that can improve the cartilage tribology. Furthermore, a range of biomimetic scaffolds, mesenchymal stem cell-based therapies, and hyaluronic acid-based gels have demonstrated promising outcomes in slowing the damage progression. Therefore, future research should prioritize enhancing the effectiveness of these techniques, as they have the potential to significantly improve the quality of life for patients affected by this degenerative joint disease.

## Figures and Tables

**Figure 1 bioengineering-11-00541-f001:**
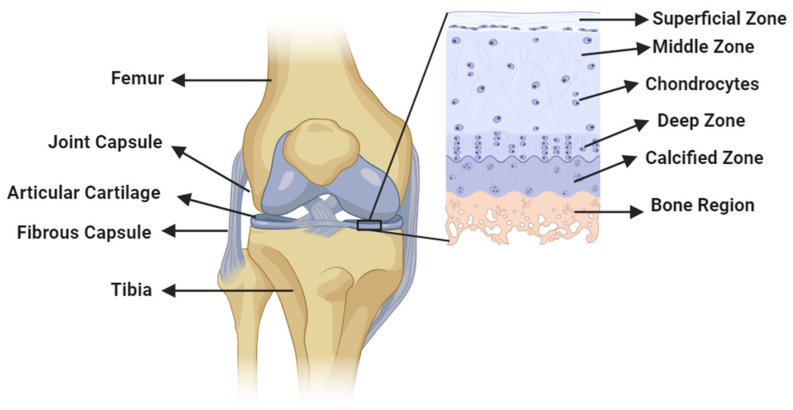
Diagram illustrating a synovial joint featuring an AC structure alongside its zonal distribution.

**Figure 2 bioengineering-11-00541-f002:**
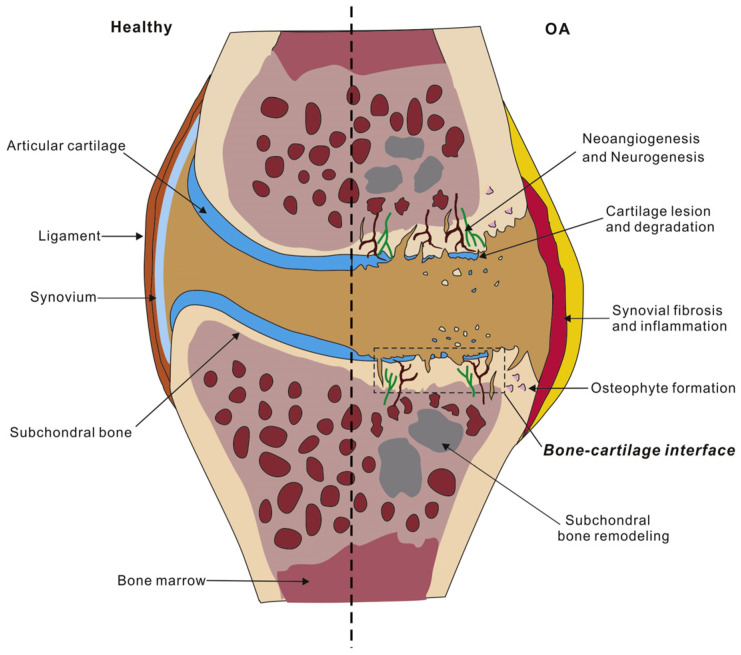
A comparison of AC in healthy and OA conditions. Reprinted with permission [31].

**Figure 3 bioengineering-11-00541-f003:**
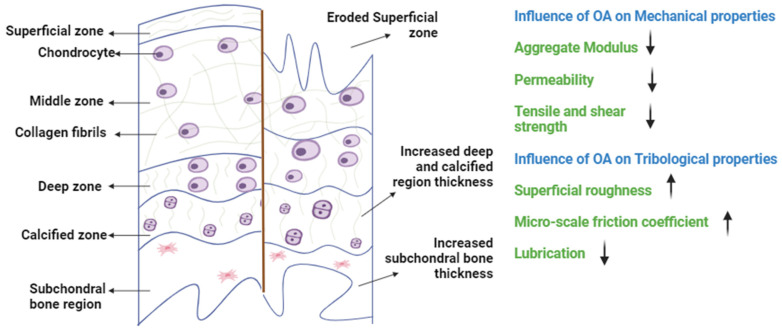
A schematic illustration of the zones of healthy and OA cartilage. In OA, the AC undergoes structural changes resulting in the loss of its mechanical and tribological properties. Specifically, the integrity of the AC is compromised as the topmost layer erodes, causing a reduction in modulus values and an increase in permeability. This deterioration further disrupts lubrication behavior, which leads to increased friction at the interface (↓ resembles decreasing, ↑ resembles increasing).

**Figure 4 bioengineering-11-00541-f004:**
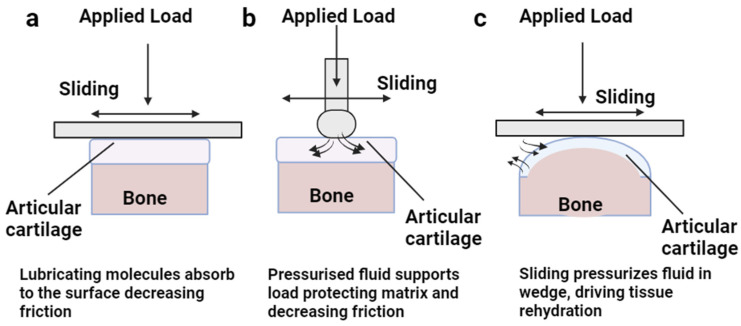
Schematic representation of three different tribological setups for AC experiments. (**a**) Stationary contact area setup, demonstrating the boundary lubrication phenomenon of AC lubrication in loaded and sliding contact with a flat surface. (**b**) Migrating contact area setup, illustrating the interstitial fluid pressurization supporting the cartilage matrix in the loaded condition using a specialized probe. (**c**) Convergent stationary contact area setup, showing the wedge-shaped cartilage surface sliding on a flat surface, enabling cartilage rehydration and providing lubrication through interstitial pressurization.

**Figure 5 bioengineering-11-00541-f005:**
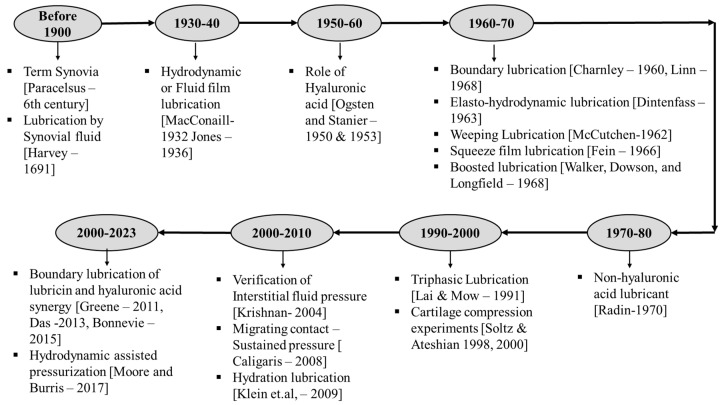
A chronological overview of AC lubrication models and significant discoveries concerning the tribological aspects of AC lubrication.

**Figure 6 bioengineering-11-00541-f006:**
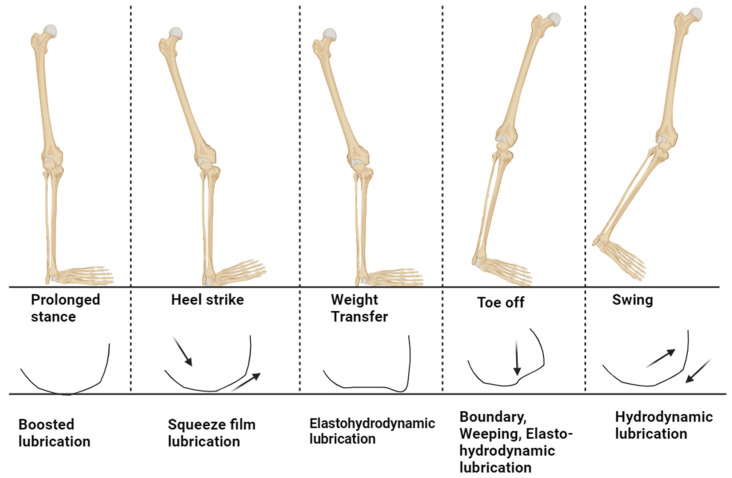
The human gait cycle, with the possible lubrication mechanism at the cartilage interface to support the load and the arrow shows the movement of fluid (recreated from reference) [114].

**Figure 7 bioengineering-11-00541-f007:**
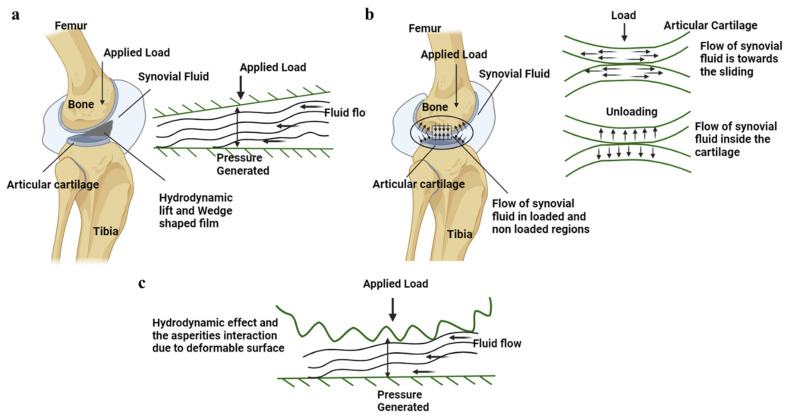
(**a**) Hydrodynamic lubrication mechanism shown at the AC interface along with the schematic representation of the pressure generated by the interstitial fluid flow. (**b**) The hydrostatic/weeping lubrication at the AC interface is shown in the figure, with a schematic representation of the flow of interstitial fluid load support during the loading and unloading cycles. (**c**) The interaction of the asperities at the AC surface with the surface is shown in the figure, and pressure is being generated at these interfaces due to the hydrodynamic lubrication. The arrow shows the fluid movement and applied load.

**Figure 8 bioengineering-11-00541-f008:**
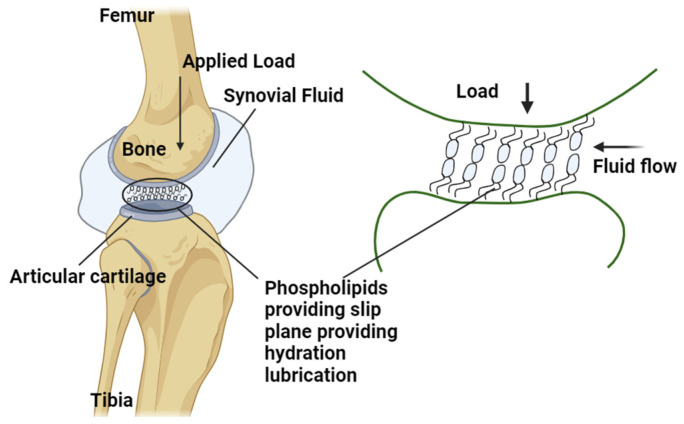
The slip plane is formed by the phospholipids acting as the boundary lubricants at the AC interface. This acts as a film supporting the load and helps provide better lubrication. The arrow shows the fluid movement and applied load.

**Figure 9 bioengineering-11-00541-f009:**
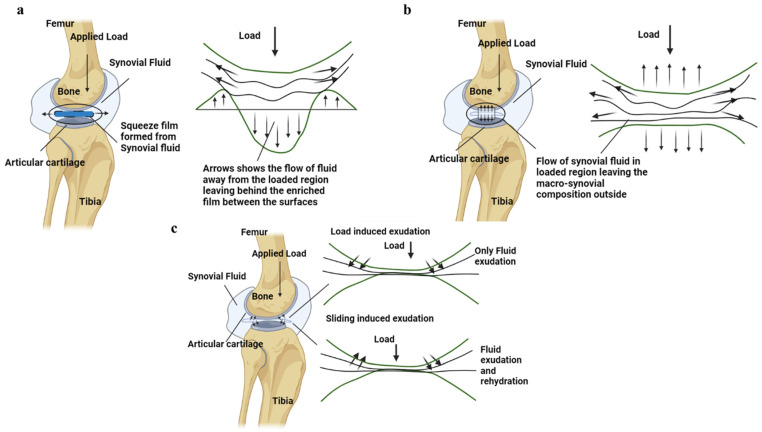
(**a**) The squeeze film formation at the interface of the AC is shown alongside the motion of fluid in and out of the loaded and unloaded region. (**b**) The fluid load support due to boosted lubrication is shown in the AC interface of the synovial joint. (**c**) The biphasic nature of the AC is shown in the figure, with the fluid flowing in and out at the AC interface during loading and sliding motions. The arrow shows the fluid movement and applied load.

**Figure 10 bioengineering-11-00541-f010:**
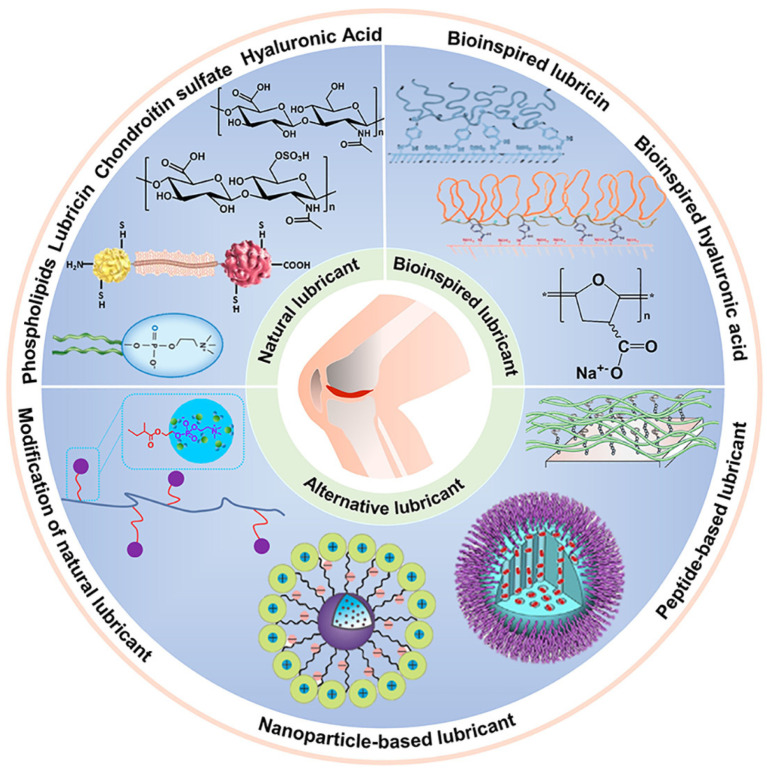
Schematic diagram showing the overview of natural and bioinspired and alternative lubricants for enhancing and restoring AC lubrication. Natural lubricants are long-chain polymeric molecules such as hyaluronic acid, chondroitin sulphate, phospholipids, and lubricin (reprinted with permission [207]).

**Table 2 bioengineering-11-00541-t002:** Summary of tribological experiments performed on AC of different species along with the details on the experimental setup and contact types used.

Articular Cartilage	Reference	Healthy/Osteoarthritic	Natural OA/Induced OA	Cartilage Region	Type of Contact	Tribological Properties			
						COF	Lubricant	Lubrication Mechanism	Other Findings
Bovine	[49]	H	-	Knee joint	MCA (stainless steel ball on cartilage)	0.024 ± 0.004	PBS	Not discussed	Fluid load fraction 0.81 ± 0.03
	[77]	H	-	Knee joint	SCA (cartilage on glass)	PBS (0.218 ± 0.015), equine SF (0.071 ± 0.012), bovine SF (0.068 ± 0.013)	PBS, bovine SF, and equine SF	Biphasic lubrication was observed along with the mixed mode and boundary lubrications; however, full-film lubrication was not observed even at high speeds.	-
	[74]	H	-	Knee joint	MCA (spherical glass lens on cartilage)	PBS (0.022 ± 0.010), SF (0.015 ± 0.004)	PBS and SF	Boundary lubrication is prominent when the thickness between the interface is lower. Fluid film lubrication is prominent when thickness is higher.	-
	[78]	H	-	Knee joint	cSCA (cartilage plug on glass slide)	0.011 ± 0.007	PBS	Tribological rehydration due to the formation of wedges, which supports full-film lubrication.	-
	[79]	H	-	Knee joint	MCA (stainless steel probe on cartilage)	0.0272 ± 0.0006–0.1168 ± 0.0014 (3.2 mm radius probe) 0.0251 ± 0.0006–0.1337 ± 0.0016 (0.8 mm radius probe)	PBS	Lubrication due to fluid pressurization	-
	[80]	OA	Induced with chondroitinase ABC and collagenase III	Knee joint	SCA (cartilage on glass)	Collagenase III (0.17 ± 0.04) and chondroitinase ABC (0.28 ± 0.02)	PBS	Biphasic behavior	-
	[81]	OA	Induced with chondroitinase ABC	Knee joint	MCA (glass on cartilage)	Chondroitinase ABC (0.19 ± 0.02)	PBS	Time dependent interstitial pressurization	-
Porcine	[82]	H	-	Knee	SCA (cartilage on glass)	0.001–0.11	SF	Weeping lubrication	-
	[83]	H	-	Knee joint	MCA (glass on cartilage)	0.04–0.14	PBS	Not discussed	-
	[84]	H	-	Knee joint	SCA (cartilage on glass)	0.039 ± 0.017–0.069 ± 0.045	PBS	Not discussed	-
	[85]	OA	Induced with hyaluronidase, Chondroitinase ABC, alkaline protease	Knee joint	SCA (cartilage on glass)	0.0025 ± 0.0012 (hyaluronidase), 0.0043 ± 0.0013 (chondroitinase ABC), 0.0070 ± 0.0003 (alkaline protease)	Normal saline	Boundary lubrication is possible due to the presence of various molecules on the surface of the cartilage.	-
Human	[86]	H	-	Knee joint	SCA (cartilage on glass)	0.22	PBS	Not discussed	-
	[87]	OA	Total joint replacement	Knee joint	MCA (cartilage on cartilage) and SCA (cartilage on glass)	MCA SF (0.019–0.02) MCA PBS (0.025–0.027) SCA SF (0.04) SCA PBS (0.09–0.12)	PBS and SF	SF lubricates better than PBS in both lesser and worse OA conditions due to its boundary lubrication properties.	-
	[80]	OA	Total joint replacement	Knee joint	SCA (cartilage on glass)	0.22 ± 0.01 (patient 1) and 0.23 ± 0.01 (patient 2)	PBS	Biphasic behavior	-
	[88]	OA	Total joint replacement	Knee joint	AFM (polysterene spherical tip on cartilage)	0.119 ± 0.036 for stage 0 (normal cartilage),0.151 ± 0.039 for stage 1, 0.158 ± 0.041 for stage 2, and 0.409 ± 0.119 for stage 3	PBS	Not discussed	Surface roughness 137 ± 25 nm for stage 0 to 533 ± 196 nm for stage 3

(H—healthy, OA—osteoarthritis, SF—synovial fluid, COF—coefficient of friction, SCA—stationary contact area, MCA—migratory contact area, cSCA—convergent stationary contact area, AFM—atomic force microscopy, PBS—phosphate buffered saline).

**Table 3 bioengineering-11-00541-t003:** The summary of lubrication models for AC, along with their physiological significance, encompassing diverse physical factors, experimental configurations, and samples employed.

Conventional Lubrication Model	Cartilage Lubrication Model	Physical Considerations	Samples	Experimental Condition	Physiological Relevance
Fluid film lubrication model	Hydrodynamic lubrication	Occurs at high articulating speeds or low load	Horse stifle joint [115] Proximal interphalangeal joint of human finger [116]	Cartilage-on-cartilage experiment [115] Modified Stanton Pendulum [116]	Swinging phase of walking and running in human gait cycle
	Hydrostatic/weeping lubrication	Occurs at constant load over time	Closed-cell rubber foam soaked with soapy water [117] Ovine AC [118] Bovine AC [24]	Pin on plate (rubber on flat surface) [117] Cartilage on glass [118] Cartilage on cartilage [24]	Stance phase of walking and running in human gait cycle
	Elastohydrodynamic lubrication	Occurs at high contact pressures and elastic deformation of AC	Human ankle joint [119] Soft material rubber [120]	Joint simulators [119] Roller bearing and soft surface [120]	Weight transfer phase due to walking, running, or jumping in human gait cycle
	Micro-elastohydrodynamic lubrication	Occurs at the microscale interaction of AC and SF. Influenced due to change in surface topography, contact deformation, and load-bearing capacity.	Human ankle joint [121]	Joint simulator [121]	During heel strike, midstance, and toe-off of the human gait cycle.
	Tribological rehydration	Modified version of hydrodynamic lubrication explaining the movements of SF into AC matrix during pressure distribution.	Bovine AC [78] Bovine, equine, porcine, ovine, and caprine [122,123]	Cartilage on flat surface [78,122,123]	Different phases of human gait cycle such as heel strike to toe-off, loading, unloading, and variable loading phases.
Boundary lubrication model	Boundary lubrication	This model considers the synovial constituents such as hyaluronic acid, lubricin, and glycoproteins.	Human knee joint [124] Human and bovine SF [125,126,127]	Modified flat-on-plate setup [124] Rheological properties of lubricin in SF [125] Pendulum oscillation in different SF concentrations [126] Hyaluronic acid rheology and concentration in SF [127]	It occurs mainly in the toe-off of the stance phase and other intermediate phases in the human gait cycle.
	Hydration lubrication	This model is an extension of boundary lubrication where it focuses mainly on the water molecules trapped inside the phospholipid layers of the synovial constituents.	Mica layers [128]	Surface force balance measurements [128]	It occurs in mainly in the toe-off of the stance phase and other intermediate phases in the human gait cycle.
Mixed lubrication model	Osmotic lubrication	Osmotic pressure gradients within cartilage matrix and interstitial fluid contributes to lubrication	Theory [129]	Theory [129]	It occurs in all the phases of human gait cycles, like the stance phase (heel strike to toe-off), the swing phase, transition phases, and dynamic movements.
	Squeeze film lubrication	Occurs when the joints are compressed, leading to interstitial fluid expulsion and redistribution and causing hydrodynamic pressure.	Glass lens with polymethylmethacrylate flats [99]	Cylinder on flat surface [99]	It occurs in the weight-bearing and relaxing phases of human gait cycles, such as heel strike and intermittent contact phases.
	Boosted lubrication	This occurs with the combination of both squeeze film and boundary lubrication.	Mathematical model [130]	Mathematical model [130]	It occurs in prolonged stances of the human gait.
	Biphasic lubrication	This considers cartilage with solid and fluid matrix and explains the load support in both strain and compressive forces.	Bovine AC [75,131,132,133]	Cartilage on metal (pin on plate) [75,131] Cartilage indentation with flat surface [132] Confined and unconfined compression [133]	It occurs in all the gait cycles of human movements.
	Triphasic lubrication	This considers the electrostatic interactions introducing an ion phase to biphasic lubrication.	Models [134,135]	Models [134,135]	It occurs in all the gait cycles of human movements.

**Table 4 bioengineering-11-00541-t004:** Overview of lubricant-based solutions for enhancing AC degeneration by utilizing modified synovial constituents.

Natural Synovial Constituent	Reference	Products/Molecular Composition	Type of Contact and Testing Apparatus	Lubricant Properties	Frictional Properties—Dynamic COF	Dose	Comments
Hyaluronic Acid	[213,214]	Synvisc One	Universal mechanical tester—Bruker (reciprocating test)	Dynamic viscosity—325.8 ± 3.4 Pa s Molecular weight 6000 kDa	0.008–0.009	Injections every 3 weeks (8 mg/mL) (2 mL)	Boundary lubrication is observed
	[214,215]	Eurflexxa	Custom tribometer (cartilage against glass sliding)	Dynamic viscosity—100.09 Pa s Molecular weight 2400–3600 kDa	0.22–0.23	Injections every 3 weeks (10 mg/mL) (2 mL)	Adsorption of molecules on the surface increased the viscosities and hence improved frictional properties
	[214,215]	Supartz	Custom tribometer (cartilage against glass sliding)	Dynamic viscosity—2.11 Pa s Molecular weight 620–1170 kDa	0.25	Injections every 5 weeks (10 mg/mL) (2.5 mL)	-
	[214]	Durolane		Molecular weight—100,000 kDa	-	1 injection (20 mg/mL) (3 mL)	-
Lubricin	[216]	mLub	Cartilage on glass surface sliding	Molecular weight ~107 kDa	0.15	-	Reduces friction and adhesion resulting in decreased cartilage degradation
	[217,218]	Proteoglycan 4 (Prg4)	Pendulum system	-	0.01	1 injection every month (250 µg/mL–10 mg/mL) (1–2 mL)	Improves chondrocytes health and prevents stick-slip at the superficial zone reducing mechanical strain and avoiding cartilage degeneration
Chondroitin Sulphate	[219]	PBS + 100 mg/mL Chondroitin sulphate	Custom designed sliding test (glass on cartilage)	-	0.05	-	Higher concentration chondroitin sulphate can improve frictional behavior at the cartilage interface
Phospholipids	[220]	Mica coated with aminothiol or poly-lysine	Surface force apparatus	-	0.08–0.3	-	The type of adsorption of the phospholipids on the surface determines how effective the frictional behavior

(COF—Coefficient of friction).

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
