# Peer review of "How Do Cartilage Lubrication Mechanisms Fail in Osteoarthritis? A Comprehensive Review"

_bioengineering, 2024, doi:10.3390/bioengineering11060541_

Round 1
Reviewer 1 Report
Comments and Suggestions for Authors
Thank you for your submission. This review discusses the characteristics of cartilage degeneration in osteoarthritis, particularly focusing on the mechanical and tribological properties of articular cartilage and the role of lubrication failure in its degradation. It explores the structure-function relationship of degraded cartilage and suggests strategies to enhance lubrication efficiency, aiming to improve the quality of life in aging populations. This manuscript aligns well with the scope of Bioengineering. However, there are several issues that need to be addressed by the authors before acceptance.
1. To enhance the readability and accessibility of the manuscript, authors should incorporate an abbreviations section at the beginning of the document.
2. The conclusion of the review would benefit from including a forward-looking perspective. Please add a discussion on promising approaches such as advanced biomaterials, novel drug delivery systems, or molecular interventions. This inclusion would provide a valuable guide for future research in this area.
3. In Section 3.1, where the manuscript discusses the possibility of combining two or three lubrication mechanisms to enhance effectiveness, it is suggested that the authors provide specific examples to substantiate this proposal.
4. In the introduction, it would be useful for the authors to include some timely reviews to provide a more comprehensive background of osteoarthritis, such as 10.3877/cma.j.issn.2096-112X.2020.01.002,and 10.12336/biomatertransl.2022.03.002. There are still several literatures relevant to the topic of section 3 and conclusion, such as 10.1016/j.jot.2020.07.008, and 10.1016/j.jot.2024.01.006.
Author Response
- Comment: To enhance the readability and accessibility of the manuscript, authors should incorporate an abbreviations section at the beginning of the document.
Response: Page-1 Abbreviation is added.
- Comment: The conclusion of the review would benefit from including a forward-looking perspective. Please add a discussion on promising approaches such as advanced biomaterials, novel drug delivery systems, or molecular interventions. This inclusion would provide a valuable guide for future research in this area.
Response: Page-27 In conclusion ‘Further a range of engineering scaffolds, mesenchymal stem cell-based therapies, and hyaluronic acid-based gels have demonstrated promising outcomes in slowing the progression of damage. Therefore, future research should prioritize enhancing the effectiveness of these techniques, as they have the potential to significantly improve the quality of life for patients affected by this degenerative joint disease’ is added.
- Comment: In Section 3.1, where the manuscript discusses the possibility of combining two or three lubrication mechanisms to enhance effectiveness, it is suggested that the authors provide specific examples to substantiate this proposal.
Response: Page-24 Section-3.1 changed to Section-5.1 ‘Several studies have been conducted to accomplish this goal, employing approaches such as mesenchymal stem cell-derived therapies, hyaluronic acid gel enhanced injections in conjunction with osteochondral scaffolds to facilitate early-stage treatments of OA [208-210]’ is added.
- In the introduction, it would be useful for the authors to include some timely reviews to provide a more comprehensive background of osteoarthritis, such as 10.3877/cma.j.issn.2096-112X.2020.01.002,and10.12336/biomatertransl.2022.03.002. There are still several literatures relevant to the topic of section 3 and conclusion, such as 10.1016/j.jot.2020.07.008, and 10.1016/j.jot.2024.01.006.
Response: Page-1 The useful articles suggested have been added to the Introduction

Reviewer 2 Report
Comments and Suggestions for Authors
Journal: Bioengineering (ISSN 2306-5354)
Manuscript ID: bioengineering-2983260
Type: Review
Title: How does cartilage lubrication mechanisms fail in Osteoarthritis? A comprehensive review
Authors
Manoj Rajankunte Mahadeshwara *, MAISOON AL-JAWAD, Richard thM Hall, Hemant Pandit, Reem El-Gendy, Michael Bryant *
Lubrication mechanism at joint is an important factor for the aetiology of OA. It’s appreciable that the review addresses this critical topic. Understanding lubrication mechanisms can guide therapeutic strategies for managing OA.
I have following comments/observations about this review:
Comment 1:
It is not clear if the authors want to limit this review for the knee joint or wants to encompass other joints too. On one hand they have included other joints in Table 1 and 2 but the discussion does not cover other joints.
Comment 2:
The lubrication dynamics will be different for different joints such as the knee joint and hip joint. I suggest the authors should give a clear statement on differences between the joints it will add to the value of this work.
Comment 3:
While comparing the animal models and human joint the important difference is bipedalism that lodes the human joint more. I suggest this point be discussed in the review.
Comment 4:
While discussing knee joint, the authors largely ignored meniscus. The meniscus plays a role in lubrication of the joint. Especially in Figure 7 where the authors discussed about Hydrodynamic lubrication mechanism shown at the cartilage interface.
Comment 5:
While discussing OA progression KL grading is not considered. There are ample papers to link these two.
Comment 6:
A comment on how the lubrication dynamics would change in patients after total knee arthroplasty will further add the value of the review.
Overall, the review is bit lengthy and there is fair scope to reduce and make it deliver a crisp message. The wordage can certainly be reduced around Figure 1. Figure 4 and related discussion does not fit aptly and stands out as an offshoot and can be curtailed.
Comments on the Quality of English LanguageThere are no major issues with English.
Author Response
Comment 1: It is not clear if the authors want to limit this review for the knee joint or wants to encompass other joints too. On one hand they have included other joints in Table 1 and 2 but the discussion does not cover other joints.
Response: Page-6 The Review is mainly considering the Knee joints and hence the other joints from Table-1 were removed.
Comment 2: The lubrication dynamics will be different for different joints such as the knee joint and hip joint. I suggest the authors should give a clear statement on differences between the joints it will add to the value of this work.
Response: Page-1 In introduction ‘Synovial joints exhibit a high level of incongruity at the central surface compared to the peripheral surface, resulting in a limited contact area at the interface’ shows that these mechanisms are common in all the synovial joints (knee, shoulder and hip). However, to restrict the content, paper mainly focuses on the work performed on the knee joints only. Was added
Comment 3: While comparing the animal models and human joint the important difference is bipedalism that lodes the human joint more. I suggest this point be discussed in the review.
Response: Page-5 ‘Additionally, the unique morphology of human knee joints is influenced by bipedal locomotion that results in increased compressive forces at the medial condyle compared to the lateral side. Consequently, this increases stress in the medial region which progresses the cartilage degradation [44,45]’wasadded.
Comment 4: While discussing knee joint, the authors largely ignored meniscus. The meniscus plays a role in lubrication of the joint. Especially in Figure 7 where the authors discussed about Hydrodynamic lubrication mechanism shown at the cartilage interface.
Response: Page-17 Also, research has highlighted the critical role of a healthy meniscus in maintaining optimal fluid load support within the knee's AC. The low permeability of a healthy meniscus serves to limit fluid exudation from the AC, thereby aiding in joint protection against OA [138]. Thus, it is evident that meniscus damage in OA leads to lower fluid load support by the AC. Is added
Comment 5: While discussing OA progression KL grading is not considered. There are ample papers to link these two.
Response: Page-23 ‘However, in cases of OA, the initial cartilage damage involves surface irregularities, matrix deterioration, and progressive wear as seen in the K-L classification. These damages are progressively increasing from Grade-1 to 4 via structural changes which negatively impact the lubrication properties. Further this damages the lubrication by the interstitial fluids, leading to increased reliance on boundary lubrication at the surface.’ is added.
Comment 6: A comment on how the lubrication dynamics would change in patients after total knee arthroplasty will further add the value of the review.
Response: The overall idea of this review is to understand the lubrication mechanism and delay the total joint replacement by suggesting alternative treatments. Hence the mechanisms after total knee replacement are out of scope which gives rise to traditional lubrication mechanisms such as metal on metal, polymer on metal, etc.
The descriptions in the Figure captions have been reduced.

Reviewer 3 Report
Comments and Suggestions for Authors
The article titled How does cartilage lubrication mechanisms fail in Osteoarthritis? A comprehensive review focuses on the breakdown of mechanical and tribological properties in articular cartilage (AC), primarily attributed to lubrication failure in osteoarthritis (OA). It provides a detailed overview of developments in AC research, emphasizing the role of lubrication in cartilage degeneration. The review discusses the structural-functional relationships of AC, the progression of its degradation, and offers potential strategies for enhancing lubrication efficiency to mitigate OA effects.
While comprehensive in its coverage of the subject, the article could benefit from more empirical data and case studies to strengthen the practical implications of its recommendations. This review is valuable for researchers and clinicians in the field of osteoarthritis, offering significant insights and contributing to the improvement of patient outcomes in OA.
The work is very extensive but generally written correctly. In some places it will require corrections and additions to the literature. I recommend minor revisions before further processing and acceptance for publication. I have included detailed comments below.
Minor comments:
The incidence of osteoarthritis is influenced by many factors, such as work, sports participation, musculoskeletal injuries, obesity and gender. Information about this, along with the necessary literature, should be added in the introduction preferably in the 1st paragraph. Authors may find some useful information in the works: DOI 10.3390/app11041552; DOI 10.3390/app10238312; doi:10.35784/acs-2022-14;
As the second paragraph of the introduction, a description should be added that includes information on typical diagnostic methods, such as computed tomography, X-ray, ultrasonography, including physical examination, as well as alternative methods such as vibroarthrography due to the fact that this method is closely related to the change of mechanical parameters of articular cartilage. The authors can find useful information in the following papers:
https://doi.org/10.1016/j.cpet.2018.08.004; https://doi.org/10.1111/j.1617-0830.2006.00063.x; DOI
10.3390/s22062176; DOI 10.3390/s22103765; https://doi.org/10.1016/j.berh.2016.09.007; DOI 10.35784/acs-2023-40.
Please be clearer about the purpose of the work and the potential of the analyses conducted, the topic undertaken is highly important.
The introduction is very long. It might be worthwhile to use a division into additional chapters instead of separating subsections in the introduction. 1.1 -> 2 Mechanical properties of AC in OA
The captions of Figures 1 and 2 contain a great deal of text. I suggest moving the content to the main text keeping only the most important information in the caption. I also recommend moving the references to the references to the main text.
The caption of Figure 10 is also too long. Please rebuild and move the information along with the references to the main text.
In summary, this article is an invaluable resource for both researchers and clinicians involved in the field of osteoarthritis and cartilage research. It successfully outlines the complexities of AC degradation and offers a clear exposition of potential therapeutic strategies, making a significant contribution to the ongoing dialogue on improving patient outcomes in OA. The clarity in writing and the structured review of existing literature make it an exemplary piece for academic and clinical reference.
After making the indicated corrections and additions, as well as supplementing with current literature, the work can be accepted for publication.
Author Response
- Comment: The incidence of osteoarthritis is influenced by many factors, such as work, sports participation, musculoskeletal injuries, obesity and gender. Information about this, along with the necessary literature, should be added in the introduction preferably in the 1st paragraph. Authors may find some useful information in the works: DOI 10.3390/app11041552; DOI 10.3390/app10238312; doi:10.35784/acs-2022-14;
Response: Page-1 Based on Reviewers suggestion ‘Further there are mechanical factors that contribute to the development of OA, including work-related activities, participation in sports, musculoskeletal injuries, and obesity [7,8]’ is added.
- Comment: As the second paragraph of the introduction, a description should be added that includes information on typical diagnostic methods, such as computed tomography, X-ray, ultrasonography, including physical examination, as well as alternative methods such as vibroarthrography due to the fact that this method is closely related to the change of mechanical parameters of articular cartilage. The authors can find useful information in the following papers: https://doi.org/10.1016/j.cpet.2018.08.004; https://doi.org/10.1111/j.1617-0830.2006.00063.x; DOI 10.3390/s22062176; DOI 10.3390/s22103765; https://doi.org/10.1016/j.berh.2016.09.007; DOI 10.35784/acs-2023-40
Response: Page-3 Based on Reviewers suggestion ‘A variety of techniques have been utilized to detect structural abnormalities resulting from OA. These include computed tomography, X-ray imaging, ultrasonography, and physical examination. Additionally, alternative methods like vibroarthrography, Raman spectroscopy, Fourier transform infrared scanning etc have been explored for this purpose [39-41]’ is added.
- Comment: The introduction is very long. It might be worthwhile to use a division into additional chapters instead of separating subsections in the introduction. 1.1 -> 2 Mechanical properties of AC in OA
Response: The Sections were Re-numbered and highlighted in the dcoument.
- The captions of Figures 1 and 2 contain a great deal of text. I suggest moving the content to the main text keeping only the most important information in the caption. I also recommend moving the references to the references to the main text. The caption of Figure 10 is also too long. Please rebuild and move the information along with the references to the main text.
Response: Page-2, The figure-1 caption is added in the main text. And in Page-3 figure-2 caption is added in the main text. Page-23 figure-10 caption is added in the main text

Reviewer 4 Report
Comments and Suggestions for Authors
The current comprehensive review discusses the importance of lubrication on joint health, the biomechanics and tests involved and lastly certain recommendations to increase lubrication efficacy. I think the authors have done a good job in summarizing the important discovery in the field and the manuscript is written well. I am looking forward to some addition on the research gaps so that the readers could get an idea how to advance this field.
Author Response
Thank you for your comments please find the modified version
